

# MagIC v5.10: a two-dimensional MPI distribution for pseudo-spectral magneto hydrodynamics simulations in spherical geometry

Rafael Lago[1], Thomas Gastine[2], Tilman Dannert[1], Markus Rampp[1], and Johannes Wicht[3]

[1]Max Planck Computing and Data Facility, Gießenbachstraße 2, 85748 Garching, Germany
[2]Institut de Physique du Globe de Paris, Université de Paris, CNRS, F-75005 Paris, France
[3]Max Planck Institute for Solar System Research, Justus-von-Liebig-Weg 3, 37077 Göttingen, Germany

**Correspondence:** Rafael Lago (rafael.lago@mpcdf.mpg.de)

**Abstract.** We discuss two parallelization schemes for MagIC, an open-source, high-performance, pseudo-spectral code for the numerical solution of the magneto hydrodynamics equations in a rotating spherical shell. MagIC calculates the non-linear terms on a numerical grid in spherical coordinates while the time step updates are performed on radial grid points with a spherical harmonic representation of the lateral directions. Several transforms are required to switch between the different representa-

tions. The established hybrid implementation of MagIC uses MPI-parallelization in radius and relies on existing fast spherical transforms using OpenMP. Our new two-dimensional MPI decomposition implementation also distributes the latitudes or the azimuthal wavenumbers across the available MPI tasks/compute cores. We discuss several non-trivial algorithmic optimizations and the different data distribution layouts employed by our scheme. In particular, the two-dimensional distribution data layout yields a code that strongly scales well beyond the limit of the current one-dimensional distribution. We also show that

the two-dimensional distribution implementation, although not yet fully optimized, can already be faster than the existing finely optimized hybrid implementation when using many thousands of CPU cores. Our analysis indicates that the two-dimensional distribution variant can be further optimized to also surpass the performance of the one-dimensional distribution for a few thousand cores.

## 1 Introduction

The dynamics in many astrophysical objects like stars, planets, or moons is aptly modelled by the fluid flow and magnetic field generation in a rotating sphere or spherical shell. Since the pioneering work by Glatzmaier (1984), several numerical codes have been developed over the years to model the problem. Typically, they all solve for convection and magnetic field induction in a co-rotating reference frame. The solutions are formulated as disturbances about a hydrostatic, well mixed, and adiabatic reference state. The so-called Boussinesq approximation assumes a homogeneous background state and yields a

particularly simple formulation. Meanwhile, the anelastic approximation (e.g. Lantz and Fan, 1999; Jones et al., 2011) allows to incorporate radial variation of the background reference state and transport properties. While the Boussinesq approximation





seems appropriate for modelling the liquid cores of terrestrial planets, the anelastic approximation is more relevant for gas planets or stars where the density contrasts are very large.

MagIC, the open source code extensively discussed here, allows to choose either Boussinesq or anelastic approximation. The initial implementation by Glatzmaier (1984), originally geared towards modelling the Solar convective zone, split in two separate codes in the 1990's. One code version was used to model the first numerical geodynamo (Glatzmaier and Roberts, 1995) and was later adopted and modified by Olson et al. (1999) to a Boussinesq code named mag. This modified code formed the basis for an earlier version of MagIC (Wicht, 2002). Stellar dynamo models were carried out using a more efficient version of the original Glatzmaier (1984) code, named ASH (Anelastic Spherical Harmonic, see Clune et al., 1999), which more recently lead to the open source code Rayleigh (Featherstone, 2018).

MagIC still mostly follows the original algorithm laid down by Glatzmaier (1984). It is a pseudo-spectral code written in modern Fortran. Pseudo-spectral algorithms use a spectral representation to compute accurate derivatives and a grid representation for calculating non-linear terms. In MagIC, the angular representation switches between a longitude/latitude grid and a spherical harmonic representation. Chebyshev polynomials are used for the spectral representation in radius, but MagIC also offers to employ finite differences instead. Different implicit-explicit time-stepping schemes (IMEX) are available, where the non-linear terms and the Coriolis force are treated explicitly, while the stiff linear terms are handled implicitly.

Over the last 20 years several aspects of the original algorithm by Glatzmaier (1984) have been tested against alternative ideas. Among them are radial discretization algorithms based on finite differences (Dormy et al., 1998), compact finite differences (Takahashi, 2012) or a sparse Chebyshev formulation (Marti et al., 2016). Various different implicit-explicit strategies for the time stepping have also been studied (e.g. Livermore, 2007; Garcia et al., 2010; Marti et al., 2016).

For the last 20 years, MagIC simulations resulted in more than 120 peer-reviewed publications[1]. They cover a broad range of scientific fields, from fundamental fluid dynamics in spherical geometry (e.g. Wicht, 2014; Barik et al., 2018), numerical modelling of the geodynamo (e.g. Wicht, 2002), and modelling of planetary (e.g. Gastine and Wicht, 2012; Heimpel et al., 2016) and stellar (e.g. Gastine et al., 2014; Raynaud et al., 2020) internal dynamics. The code has been tested and validated by community-driven comparison benchmarks for Boussinesq convection (Christensen et al., 2001), anelastic convection (Jones et al., 2011), double-diffusive convection (Breuer et al., 2010), and convection in full spheres (Marti et al., 2014). Several publications concern the study of the solid Earth and its magnetic field. For instance, MagIC was used to study the "top-heavy" regime of double-diffusive convection, when thermal and compositional background gradients are destabilizing (Tassin et al., 2021). In a recent paper, MagIC has been used to study magnetic boundary layers with heterogeneous outer boundary heat flux, with results suggesting a significant deviation from classical estimates of the diffusion time and the magnetic Reynold number (Terra-Nova and Amit, 2020). In another paper, the dynamics of a possible stable stratification layer atop Earth's core was explored (e.g. Gastine et al., 2020), leading to the conclusion that such a layer would lead to strong magnetic skin effects, incompatible with current observations.

Achieving an efficient use of the available computer resources by a given numerical implementation often proves challenging, in particular when moving to petascale architectures. There are two main reasons. First, the large number of physical

---

[1]See https://ui.adsabs.harvard.edu/user/libraries/LVt1vdaKQsC5P09In2iloA.





compute cores requires a large enough workload that can be fairly partitioned and distributed across these processing units. This typically puts an upper bound on the number cores that can usefully be employed and often results in poor strong scaling. A second complication arises from the different layers of memory access and of communication between the physical cores (i.e. NUMA domains, sockets, nodes etc). Ideally, one would attempt to keep all data "local" for quick access. However, opti-

mizing a code for properly distributing the workload while keeping a reasonable data locality can be difficult and often requires compromising one aspect in favour of another.

Until recently, MagIC only offered a one-dimensional distribution of the data, implemented using MPI+OpenMP (hereafter referred to as *"1d-hybrid"* implementation). For calculating the non-linear terms in grid space, this code uses MPI to distribute the spherical shells between the available NUMA domains and the inherent OpenMP scheme of the open-source spherical

harmonics transform library SHTns (Schaeffer, 2013) for computations within a NUMA domain. Since MagIC is not optimized for multithreading between NUMA domains, this configures a limitation on the maximum number of the radial grid points. This limitation could be quite severe. HPC clusters typically have been offering two NUMA domains per compute node, but the current trend in hardware development might lead into a further subdivision into multiple physical units (e.g. AMD EPYC and Intel CascadeLake-AP) or logical NUMA domains (e.g. sub-NUMA clustering in Intel Xeon). Effectively, the computing

power per NUMA domain has stagnated and it may even decrease in the future. This would simply prevent MagIC from taking advantage of performance gains in the foreseeable evolution of processor technology. In addition, numerical experiments Matsui et al. (2016) suggest that two- or three-dimensional parallelization might be necessary for dynamo codes to achieve petascale capabilities with current hardware architecture.

The first two-dimensional MPI domain decomposition in pseudo-spectral codes in spherical geometry was considered in

the ASH code by Clune et al. (1999). The layout was nevertheless still not completely two-dimensional in the sense that the maximum number of MPI ranks was still bounded by the maximum spherical harmonic degree. This limitation was released in more recent implementations by Marti and Jackson (2016) in the context of modelling full spheres, by Matsui et al. (2014) in the geodynamo code Calypso and in the Rayleigh code. The performance benchmark by Matsui et al. (2016) however revealed an important overhead for these codes compared to pseudo-spectral codes which avoid global communications such as the

open source code XSHELLS (e.g. Schaeffer et al., 2017).

Motivated by the aforementioned points, we propose in this work a two-dimensional data distribution layout for MagIC with communication-avoiding features. This required a major rethinking of data structures and communication algorithms and demands a re-implementation and thorough optimization of a large portion of the existing 1d-hybrid code. Due to the high complexity of the required refactoring tasks, OpenMP parallelism was dropped from the current implementation for the time

being. Since the two-dimensional distribution implementation presented here relies on pure MPI communication, we refer to it as the *"2d-MPI"* implementation or version. Implications and incentives to eventually re-introduce OpenMP into the new version will be discussed along the way.

This paper is organized as follows. In section 2 we provide an overview of the mathematical formulation and the numerical techniques used in MagIC. In section 3 we detail the pseudo-codes for the sequential algorithm, for the 1d-hybrid and the

2d-MPI implementations. We also describe the underlying data structures and the key differences between each approach. In





section 4 we thoroughly compare the different parts of 1d-hybrid and 2d-MPI implementations. We focus our discussion on the strong scaling of both algorithms, but we also discuss the raw performance in different regimes. Especially in subsection 4.5 we show the performance and parallel efficiency of both implementations. In subsection 4.6 we focus on analysing the main bottleneck of the 2d-MPI implementation and discuss the viability of further performance improvements and their expected impact. Finally, in section 5 we summarize our findings.

## 2 Magneto hydrodynamics equations and numerical formulation

In this section we introduce aspects of the numerical formulation which are relevant for the understanding of this work. Since the implementation of the magneto hydrodynamic (or MHD for short) equations implemented in MagIC still closely follows the original work by Glatzmaier (1984), we refer to this publication and to more recent reviews (e.g. Tilgner, 1999; Hollerbach, 2000; Christensen and Wicht, 2015) for a more comprehensive description of the algorithm.

### 2.1 Dynamo model

We consider a spherical shell of inner radius $r_i$ and outer radius $r_o$ rotating about the $z$-axis with a constant frequency $\Omega$. Convective motions are driven by a fixed temperature contrast $\Delta T$ between both boundaries. We adopt a dimensionless formulation of the MHD equations using the shell gap $d = r_o - r_i$ as the reference length scale and the viscous diffusion time $d^2/\nu$ as time unit. Velocity is dimensionlessly represented in units $\nu/d$, temperature in units $\Delta T$, and the magnetic field in units $\sqrt{\rho\mu\lambda\Omega}$, where $\mu$ is the magnetic permeability, $\nu$ the kinematic viscosity, and $\lambda$ the magnetic diffusivity. The dimensionless equations that control the time evolution of the velocity $\boldsymbol{u}$, the magnetic field $\boldsymbol{B}$, and the temperature $T$ under the Boussinesq approximation read

$$\boldsymbol{\nabla} \cdot \boldsymbol{u} = 0, \quad \boldsymbol{\nabla} \cdot \boldsymbol{B} = 0, \tag{1}$$

$$\frac{\partial \boldsymbol{u}}{\partial t} + \boldsymbol{u} \cdot \boldsymbol{\nabla}\boldsymbol{u} + \frac{2}{E}\boldsymbol{e_z} \times \boldsymbol{u} = -\nabla p + \frac{Ra}{Pr}gT\,\boldsymbol{e_r} + \frac{1}{E\,Pm}\left(\boldsymbol{\nabla} \times \boldsymbol{B}\right) \times \boldsymbol{B} + \nabla^2 \boldsymbol{u}, \tag{2}$$

$$\frac{\partial \boldsymbol{B}}{\partial t} = \boldsymbol{\nabla} \times \left(\boldsymbol{u} \times \boldsymbol{B}\right) + \frac{1}{Pm}\nabla^2 \boldsymbol{B}, \tag{3}$$

$$\frac{\partial T}{\partial t} + \boldsymbol{u} \cdot \boldsymbol{\nabla}T = \frac{1}{Pr}\nabla^2 T, \tag{4}$$

where $p$ is the non-hydrostatic pressure, $\boldsymbol{e_r}$ and $\boldsymbol{e_z}$ are the unit vectors along the radial and axial directions respectively, and $g = r/r_o$ is the dimensionless gravity. This set of equations is controlled by four dimensionless parameters, the Rayleigh number $\mathrm{Ra}$, the Ekman number $\mathrm{E}$, the magnetic Prandtl number $\mathrm{Pm}$, and the Prandtl number $\mathrm{Pr}$

$$E = \frac{\nu}{\Omega d^2}, \, Ra = \frac{\alpha g_o d^3 \Delta T}{\nu\kappa}, \, Pr = \frac{\nu}{\kappa}, \, Pm = \frac{\nu}{\lambda}, \tag{5}$$





where $\alpha$ is the thermal expansivity, $g_o$ is the gravity at the outer boundary, and $\kappa$ is the thermal diffusivity. Equations (1)–(4) need to be complemented by appropriate boundary conditions on temperature, velocity, and magnetic field as well as an initial state.

## 2.2 Spatial discretization

To ensure the solenoidal nature of $\boldsymbol{u}$ and $\boldsymbol{B}$, Eq. (2)–(4) are solved in the spherical coordinates $(r,\theta,\phi)$ by expanding the

velocity and the magnetic fields into poloidal and toroidal potentials following

$$\boldsymbol{u} = \boldsymbol{\nabla} \times (\boldsymbol{\nabla} \times W \boldsymbol{e_r}) + \boldsymbol{\nabla} \times Z \boldsymbol{e_r},$$

$$\boldsymbol{B} = \boldsymbol{\nabla} \times (\boldsymbol{\nabla} \times G \boldsymbol{e_r}) + \boldsymbol{\nabla} \times H \boldsymbol{e_r}.$$

This formulation yields the six time-dependent scalar quantities $W$, $Z$, $G$, $H$, $T$, and $p$. These six variables are expanded in spherical harmonic functions up to a degree and order $\ell_{\max}$ in the angular directions $(\theta,\phi)$. Since the spherical harmonics are the set of eigenfunctions of the Laplace operator on the unit sphere, they are an especially attractive basis for representing

functions in spherical coordinates. They are defined by

$$Y_{\ell m}(\theta,\phi) = P_{\ell m}(\cos\theta)e^{im\phi}, \tag{6}$$

where $P_{\ell m}$ are the normalized Legendre polynomials (see Abramowitz and Stegun, 1964). Any function $f(r,\theta,\phi,t)$ at the radius $r$ and instant $t$ can be expanded by a truncated spherical harmonic decomposition

$$f(r,\theta,\phi,t) \approx \sum_{\ell=0}^{\ell_{\max}} \sum_{m=-\ell}^{\ell} f_{\ell m}(r,t) Y_{\ell m}(\theta,\phi). \tag{7}$$

Letting $\star$ denotes the complex conjugate, it holds that $f_{\ell,-m}(r,t) = f_{\ell m}^{\star}(r,t)$ for all real functions $f$. This effectively halves the cost of computing and storing the spherical harmonic coefficients.

Reordering the terms in Eq. (7) and using the definition of the spherical harmonics from Eq. (6) we obtain

$$f(r,\theta,\phi,t) \approx \sum_{m=-\ell_{\max}}^{\ell_{\max}} f_m(r,\theta,t)\, e^{im\phi}, \qquad \text{(Fourier transform)} \tag{8}$$

$$f_m(r,\theta,t) \approx \sum_{\ell=|m|}^{\ell_{\max}} f_{\ell m}(r,t) P_{\ell m}(\cos\theta), \qquad \text{(Legendre transform)} \tag{9}$$

which is known as the inverse spherical harmonics transform. By integrating $f_{\ell m}Y_{\ell m}$ over the spherical coordinates using the Gauss-Legendre quadrature we obtain the forward transform:

$$f_m(r,\theta,t) = \frac{1}{2\pi}\int_0^{2\pi} f(r,\theta,\phi,t)e^{-im\phi}\,\mathrm{d}\phi \approx \frac{1}{N_\phi}\sum_{k=1}^{N_\phi} f(r,\theta,\phi_k,t)e^{-im\phi_k}, \qquad \text{(Fourier transform)} \tag{10}$$

$$f_{\ell m}(r,t) = \frac{1}{\pi}\int_0^{\pi} f_m(r,\theta,\phi,t)P_{\ell m}(\cos\theta)\sin\theta\,\mathrm{d}\theta \approx \frac{1}{N_\theta}\sum_{j=1}^{N_\theta} w_j f_m(r,\theta_j,t)P_{\ell m}(\cos\theta_j), \qquad \text{(Legendre transform)} \tag{11}$$



where $\theta_j$ and $w_j$ are respectively the Gauss nodes and weights for the $N_\theta$ collocation points in the latitudinal direction and
$\phi_k = 2k\pi/N_\phi$ are the $N_\phi$ regularly-spaced grid points in the azimuthal direction.

The formulation discussed in Eq. (8)–(11) provides a two-step algorithm for computing the spherical harmonics transforms (hereafter SHT), corresponding to a Legendre transform and a Fourier transform. The latter can be handled by finely optimized implementations of the fast Fourier transform (FFT for short) such as the FFTW library (Frigo and Johnson, 2005), which costs $\mathcal{O}(N_\phi \log N_\phi)$ operations. There is no fast transform for Eq. (11), and the complexity of an individual Legendre transform scales as $\mathcal{O}(N_\theta^2)$ operations. In MagIC, we use the open-source library SHTns for SHTs and Legendre transforms (Schaeffer, 2013), which relies on the recurrence relation by Ishioka (2018). Additionally, in the context of practical numerical implementation, a transposition is required between the two transforms; this will be covered further in section 3.

The quadrature shown in Eq. (11) is exact for $N_\theta > \ell_{\max}$ (Schaeffer, 2013), but in practice, we set $\ell_{\max} = \lceil 2N_\theta/3 \rceil$ in order to prevent aliasing errors (e.g. Orszag, 1971; Boyd, 2001). In fact, we also set $N_\phi = 2N_\theta$ for the number of longitudinal points, which guarantees isotropic resolution in the equatorial regions (Glatzmaier, 2013) .

During the initialization stage, MagIC allows the choice between finite differences or a spectral expansion using Chebyshev polynomials to handle the radial discretization strategy. Finite differences method allows the use of faster point to point communications but they also require a larger number of nodal points to ensure a proper convergence of the solution. In this work, we explicitly chose to focus solely on the spectral approach to handle the radial discretization, but we encourage the interested reader to consult (Matsui et al., 2016) for an extensive comparison between several different numerical implementations.

Each spectral coefficients $f_{\ell m}(r, t)$ is expanded in truncated Chebyshev series up to the degree $N_c - 1$

$$f_{\ell m}(r_k, t) \approx c \sum_{n=0}^{N_c-1}{}'' f_{\ell m n}(t)\, C_n[x(r_k)],\tag{12}$$

where $c = \sqrt{2/N_r - 1}$ is a normalization factor and the double quotes mean that the first and last terms need to be multiplied by one half. In the above expression, $C_n(x)$ is the $n$th order first-kind Chebyshev polynomial defined by

$$C_n(x_k) = \cos[n \arccos(x_k)],\, x_k \in [-1, 1] \subset \mathbb{R},$$

with

$$x_k = \cos\frac{(k-1)\pi}{N_r},\, k \in [1, N_r],$$

the $k$th nodal point of a Gauss-Lobatto grid defined with $N_r$ collocation points. The discrete radius $r_k$ defined between $r_i$ and $r_o$ is mapped onto $x_k$ using the following affine mapping

$$r_k = \frac{r_o - r_i}{2} x_k + \frac{r_o + r_i}{2},\, k \in [1, N_r].$$

Conversely, the Chebyshev spectral coefficients of the functions $f_{\ell m}(r, t)$ read

$$f_{\ell m n}(t) \approx c \sum_{k=1}^{N_r}{}'' f_{\ell m}(r_k, t)\, C_n(x_k).\tag{13}$$





With the particular choice of Gauss-Lobatto collocation points, Eq. (12) and Eq. (13) can also be efficiently computed using fast discrete Cosine transforms (DCTs) of the first kind (Press et al., 2007, § 12.4.2).

## 2.3 Time discretization

With the spatial discretization fully specified, we can proceed with the time discretization. For an easier understanding, we will derive the main steps using the equation for the time evolution of the magnetic poloidal potential $G$ as an illustrative example. We refer the interested reader to (e.g. Christensen and Wicht, 2015) or the online documentation of MagIC for the derivation of the other equations.

The equation for $G$ is obtained by considering the radial component of the induction equation (see Eq. (3)). Using the spherical harmonics decomposition and the Chebyshev collocation method described above we obtain

$$\frac{\ell(\ell+1)}{r_k^2}c\sum_{n=0}^{N_c-1}\left[\left(\frac{\mathrm{d}}{\mathrm{d}t}+\frac{1}{Pm}\frac{\ell(\ell+1)}{r_k^2}\right)C_n(x_k)-\frac{1}{Pm}C_n''(x_k)\right]G_{\ell mn}(t)=\int[\boldsymbol{e_r}\cdot\boldsymbol{\nabla}\times(\boldsymbol{u}\times\boldsymbol{B})]Y_{\ell m}^\star\,\mathrm{d}\mathcal{S}, \tag{14}$$

where $C_n''$ is the second radial derivative of the $n$th Chebyshev polynomial and $\mathrm{d}\mathcal{S}=\sin\theta\,\mathrm{d}\theta\mathrm{d}\phi$ is the spherical surface element.

Equation (14) can be rewritten in the following matrix form

$$\mathbf{A}_\ell\frac{\mathrm{d}\boldsymbol{G_{\ell m}}(t)}{\mathrm{d}t}=\boldsymbol{\mathcal{N}}_{\ell m}[\boldsymbol{u},\boldsymbol{B}]+\mathbf{L}_\ell\boldsymbol{G_{\ell m}}(t), \tag{15}$$

where the matrices $\mathbf{A}_\ell$ and $\mathbf{L}_\ell\in\mathbb{R}^{N_r\times N_r}$ contain primarily the coefficients of $C_n(x_k)$ and $C_n''(x_k)$, are dense, and depend on $\ell$ but not on $m$. Equation (15) forms a set of ordinary differential equations that depend on time and contain a non-linear term, namely $\boldsymbol{\mathcal{N}}_{\ell m}(t)\in\mathbb{C}^{N_r}$, and a stiff linear diffusion operator on the right hand side.

In order to mitigate the time step constraints associated with an explicit treatment of the diffusion terms, MagIC adopts an implicit-explicit (IMEX) time stepping approach. Non-linear and Coriolis terms are handled using the explicit part of the time integrator, while the remaining linear terms are treated implicitly. Currently, IMEX multisteps (e.g. Ascher et al., 1995) or IMEX diagonally-implicit Runge-Kutta (DIRK, e.g. Ascher et al., 1997) are available.

Let $\delta t$ denote the time step size. A general $k$ step IMEX multistep method applied to Eq. (15) reads

$$(\mathbf{A}_\ell-\delta t\,b_0^{\mathcal{I}}\mathbf{L}_\ell)\boldsymbol{G_{\ell m}^{i+1}}=\sum_{j=1}^k a_j\mathbf{A}_\ell\boldsymbol{G_{\ell m}^{i+1-j}}+\delta t\sum_{j=1}^k\left(b_j^{\mathcal{E}}\boldsymbol{\mathcal{N}_{\ell m}^{i+1-j}}+b_j^{\mathcal{I}}\mathbf{L}_\ell\boldsymbol{G_{\ell m}^{i+1-j}}\right), \tag{16}$$

where the exponent notations correspond to the time discretization with $t^i=t^0+i\delta t$ and $a_j$, $b_j^{\mathcal{E}}$, and $b_j^{\mathcal{I}}$ correspond to the weights of the IMEX multistep scheme. In practice, multistep schemes present stability domains that decrease with their order of convergence and require the knowledge of past states to continue the time integration. As such, they are not self-starting and need to be initiated with a lower order scheme. MagIC implements DIRK schemes to overcome these limitations, but here we discuss only a simple two step IMEX scheme. Setting the IMEX weights to $\boldsymbol{a}=(1,0)$, $\boldsymbol{b^{\mathcal{E}}}=(3/2,-1/2)$ and $\boldsymbol{b^{\mathcal{I}}}=(1/2,1/2,0)$, Eq. (16) reduces to the popular IMEX scheme assembled from a Crank–Nicolson and a second order





Adams–Bashforth scheme (Glatzmaier, 1984) and can simply be written as

$$\underbrace{\left(\mathbf{A}_\ell - \frac{\delta t}{2}\mathbf{L}_\ell\right)}_{\mathbf{M}_\ell^{\delta t}} \boldsymbol{G}_{\ell m}^{i+1} = \underbrace{\left(\mathbf{A}_\ell + \frac{\delta t}{2}\mathbf{L}_\ell\right)\boldsymbol{G}_{\ell m}^{i} + \frac{\delta t}{2}\left(3\boldsymbol{\mathcal{N}}_{\ell m}^{i} - \boldsymbol{\mathcal{N}}_{\ell m}^{i-1}\right)}_{\boldsymbol{B}_{\ell m}^{i}}, \tag{17}$$

with $\mathbf{M}_\ell^{\delta t} \in \mathbb{R}^{N_r \times N_r}$ and $\boldsymbol{G}_{\ell m}^{i+1}, \boldsymbol{B}_{\ell m}^{i} \in \mathbb{C}^{N_r}$. Assuming suitable initial and boundaries conditions, all terms needed to compute $\boldsymbol{B}_{\ell m}^{i}$ are known from the previous iterations. The matrices $\mathbf{M}_\ell^{\delta t}$ contain the Chebyshev coefficients and are dense, but real-

valued and of moderate size (realistic values of $N_r$ range from 33 to 1025). As such, the linear system in Eq. (17) requires typically $\mathcal{O}(N_r^2)$ operations to be inverted (Boyd, 2001) and can be easily solved using LU decomposition and backward substitution from standard libraries such as LAPACK (Anderson et al., 1999). From here on, we abuse the notation and use Eq. (17) when discussing the time stepping for any fields, i.e. not only the magnetic poloidal potential $G$.

## 3 Implementation

MagIC is a highly optimized hybrid MPI+OpenMP code under active development. The 1d-hybrid version uses MPI to distribute the $N_r$ spherical shells amongst different computing nodes, while employing OpenMP for fully using the local cores within an individual NUMA domain. Although MagIC is finely optimized, multithreading across NUMA domains is not implemented. Therefore, simulations require at least one spherical shell per NUMA domain. This restricts the maximum number of NUMA domains to $N_r - 1$, which severely limits the computational resources that can be employed for a given radial res-

olution. This disadvantage could increase in the future since the current trend in HPC architecture is to further sub-divide the computing units into physical or even logical NUMA domains.

The purpose of this section is to first familiarize the reader with the established 1d-hybrid implementation and then introduce the new 2d-MPI implementation. By adding MPI-parallelism in a second direction, the extension also allows distributing the computations within a shell over the NUMA domains, sockets or even nodes of a computer cluster.

Due to the high complexity of this re-implementation, our 2d-MPI version lacks any use of OpenMP. The main purpose of this work is to provide a thorough assessment of the prospects and merits of a two-dimensional data distribution, to pinpoint shortcomings, and to discuss the overall viability of a possible fully optimized and fine-tuned two-dimensional distribution implemented using OpenMP+MPI.

In this section we first present the pseudocode for the sequential algorithm for MagIC in subsection 3.1 and then discuss the

distribution of the data both for the 1d-hybrid and 2d-MPI versions in subsection 3.2. A more detailed description of the two main parts of the code are discussed in subsection 3.3 and in subsection 3.4.

### 3.1 Sequential Algorithm

For the sake of simplicity, we discuss only the second order time stepping scheme described in subsection 2.3, Eq. (17). The resulting pseudocode is shown in Algorithm 1. For each time step, the code can be divided into two stages, the "radial loop"

and the "$\ell$-loop":





**The radial loop (lines 2–10):** It computes the non-linear terms in grid space $(\theta, \phi, r)$ and thus requires forward and inverse SHTs. The radial levels are completely decoupled in this stage, and thus, can be efficiently distributed without any need for communication between them. The directions $(\ell, m)$ and $(\theta, \phi)$ are coupled. Notice that the FFTs on lines 6 and 9 take place on the second, slowest dimension. This requires the Fourier transform to "stride" the data in memory. This is not ideal, but

modern FFT libraries are able to efficiently handle strided transforms.

**The $\ell$-loop (lines 13–18):** It performs the actual time step. Most of the computation effort goes into solving the linear systems from Eq. (17) and into updating the right hand sides $\boldsymbol{B}^i_{\ell m}$. After updating each $\boldsymbol{B}^i_{\ell m}$, the factors of $\mathbf{M}^{\delta t}_{\ell}$ are computed using LAPACK's real-valued LU decomposition `dgetrf` routines for dense matrices. Next, the solution of each linear system from Eq. (17) needs to be computed. In practice, all right hand sides are "packed" in a real matrix $\mathbf{B}^i_\ell \in \mathbb{R}^{N_r \times (2\ell+2)}$:

$$\mathbf{B}^i_\ell = \left[ \Re(\boldsymbol{B}^i_{\ell 0}), \Re(\boldsymbol{B}^i_{\ell 1}), ... \Re(\boldsymbol{B}^i_{\ell\ell}), \Im(\boldsymbol{B}^i_{\ell 0}), \Im(\boldsymbol{B}^i_{\ell 1}), ... \Im(\boldsymbol{B}^i_{\ell\ell}) \right] \tag{18}$$

where $\Re$ and $\Im$ respectively represent the real and imaginary parts of the vector. The system is then solved for all right hand sides as

$$\mathbf{M}^{\delta t}_\ell \mathbf{X} = \mathbf{B}^i_\ell \tag{19}$$

with a single call to LAPACK's `dgetrs`. The matrix $\mathbf{B}^i_\ell$ is computed and stored as $(r, m, \ell)$ for faster memory access during

the LAPACK call. Therefore, the intermediate solution $\mathbf{X}$ is also obtained as $(r, m, \ell)$. The final solution $\boldsymbol{G}^{i+1}_{\ell m}$ is reconstructed from $\mathbf{X}$ by reordering the data back into the $(\ell, m, r)$ format, while adding the real and the imaginary parts back together. Notice also that the decomposed factors of $\mathbf{M}^{\delta t}_\ell$ are kept for all subsequent time steps, and are only recomputed when $\delta t$ changes.

## 3.2   Data Distribution

In this section we discuss how the simulation is distributed across MPI ranks for the 1d-hybrid and the 2d-MPI implementations. For all purposes, MPI Cartesian grid topology is used. The $\phi$ direction is numerically periodic, but this property is never explicitly used by the underlying algorithms. The remaining directions are non-periodic. We denote by $\Pi_r$ the number of MPI ranks used in the radial direction. Similarly, $\Pi_\theta$ represents the number of ranks in $\theta$ or $\ell$ direction.

     Let $N_\ell = \ell_{\max} + 1$ denote the total number of $\ell$-modes, $N_m$ the number of $m$-modes and $N_{\ell m}$ the number of $(\ell, m)$ tuples.

We simply use the $^-$ superscript to denote the number of *local points* in some MPI rank. For instance, $N^-_r$ denotes the number of local radial points stored in a given rank.

**1d-hybrid distribution:** in the radial loop, the data distribution for the 1d-hybrid approach follows intuitively from Algorithm 1. The radial shells are distributed evenly among the MPI ranks, ideally $N^-_r = N_r/\Pi_r$ for each rank. During the computation of the non-linear terms, each rank stores $N_\phi \times N_\theta \times N^-_r$ points. During the computation of the miscellaneous terms,

each rank stores $N_{\ell m} \times N^-_r$ points.

     This changes for the execution of the $\ell$-loop. For efficiently computing line 17 of Algorithm 1, all $N_r$ radial points must be local. This is achieved by the so-called $\ell$-transposition, which changes the data distribution to $N_r \times N^-_{\ell m}$ local points. Special





---

**Algorithm 1** Pseudocode for the sequential MagIC - Crank–Nicolson with second order Adams–Bashforth scheme

---

1: **for** each time step **do**

2:   *! "radial loop": non-linear terms computation*

3:   **for** each $r$ in $N_r$ **do**

4:     *! inverse spherical harmonics transform, Eq. (10)–(11)*

5:     Legendre transform: $(\ell, m, r) \rightarrow (\theta, m, r)$

6:     FFT: $(\theta, m, r) \rightarrow (\theta, \phi, r)$

7:     compute non-linear terms

8:     *! forward spherical harmonics transform Eq. (8)—(9)*

9:     FFT: $(\theta, \phi, r) \rightarrow (\theta, m, r)$

10:     Legendre transform: $(\theta, m, r) \rightarrow (\ell, m, r)$

11:     compute misc terms *! Coriolis force*

12:   **end for**

13:   *! "$\ell$-loop": IMEX time stepping Eq. (17)*

14:   **for** each $\ell$ in $N_\ell$ **do**

15:     **for** $m = 0$ to $\ell$, update $\boldsymbol{B}_{\ell m}^i$

16:     **if** $\delta t$ changed **then** recompute: $\mathbf{M}_\ell^{\delta t} \in \mathbb{R}^{N_r \times N_r}$ and its factors

17:     **for** $m = 0$ to $\ell$, solve $\mathbf{M}_\ell^{\delta t} \boldsymbol{G}_{\ell m}^{i+1} = \boldsymbol{B}_{\ell m}^i$

18:     update radial derivatives

19:   **end for**

20: **end for**

---

care is needed when choosing the distribution of the $N_{\ell m}^-$ points. The computational effort per $\ell$-modes is given by the LU-decomposition of the matrix and the solve step for $\ell + 1$ right-hand-side vectors for this matrix. To balance the computation among the $\Pi_r$ processes, we distribute the $\ell$-modes in a "snake-like" fashion. Starting with the largest $\ell$, which has the highest computational effort, we distribute the $\ell$ modes over the processors until $\Pi_r$ is reached. The next $\ell$ is also placed on this last processor, and then we reverse the order until reaching the first processor again. The procedure is repeated until all modes have been distributed. See Fig. 1 for an illustration with $N_\ell = 21$ and $\Pi_r = 6$. This snake-like distribution guarantees that the largest

$$
\begin{array}{cccccc}
\mathbf{0} & \mathbf{1} & \mathbf{2} & \mathbf{3} & \mathbf{4} & \mathbf{5} \\
20 \rightarrow & 19 \rightarrow & 18 \rightarrow & 17 \rightarrow & 16 \rightarrow & 15 \\
 & & & & & \downarrow \\
9 \leftarrow & 10 \leftarrow & 11 \leftarrow & 12 \leftarrow & 13 \leftarrow & 14 \\
\downarrow & & & & & \\
8 \rightarrow & 7 \rightarrow & 6 \rightarrow & 5 \rightarrow & 4 \rightarrow & 3 \\
 & & & & & \downarrow \\
 & 0 \leftarrow & 1 \leftarrow & 2 & &
\end{array}
$$

**Figure 1.** Illustration of the "snake-ordering" of the $(\ell, m)$ tuples for the 1d-hybrid distribution with $N_\ell = 21$, $\Pi_r = 6$. The top row shows the processor number in boldface, the remaining numbers indicate the $\ell$-modes. To each $\ell$ value, $\ell + 1$ solution steps are associated. The arrows illustrate the snake-like winding pattern of this ordering.



possible imbalance between two MPI ranks is $\Pi_r - 1$ tuples. Furthermore, each matrix $\mathbf{B}_\ell^i$ can be fully stored locally, which is

not the case for the 2d-MPI code, as we discuss below. At the end of the $\ell$-loop, another $\ell$-transposition takes place, converting the data layout back to $N_{\ell m} \times N_r^-$ local points.

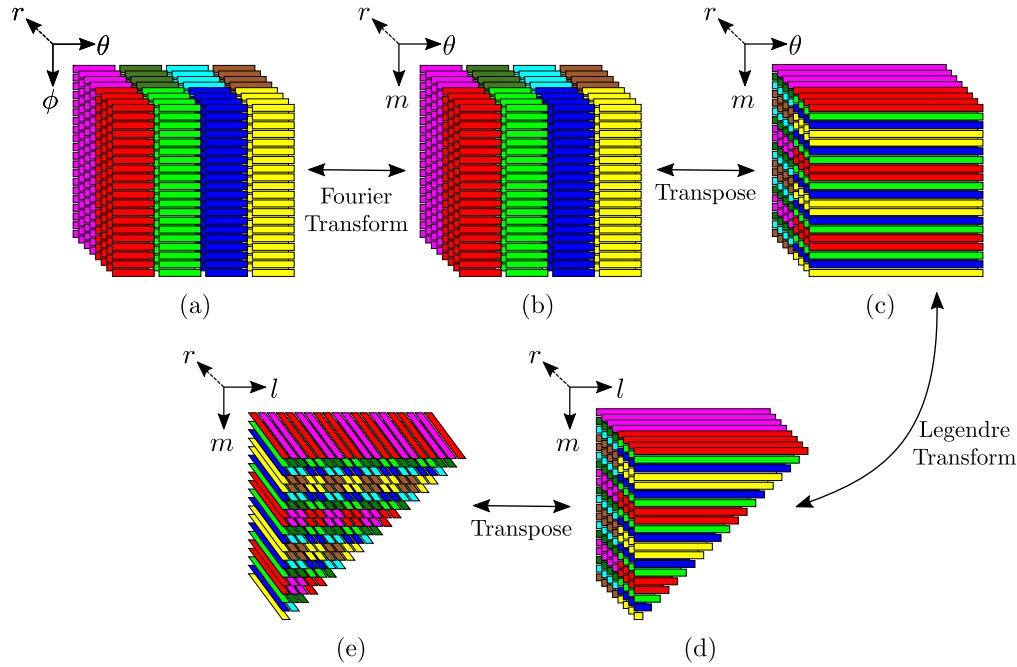

**Figure 2.** Illustration of the 2d-MPI distribution of the points with $\Pi_r = 2$, $\Pi_\theta = 4$, $N_r = 8$ and $N_\phi = N_\theta = N_\ell = 20$ (a more realistic proportion would have been $N_\phi = 2N_\theta = 3N_\ell/2$). Continuous stripes represent continuous memory segments and different colours represent distinct MPI ranks. The images show which MPI rank would store and process each data segment during (a) computation of non-linear terms, (b)–(d) forward SHT, (e) beginning of $\ell$-loop.

**2d-MPI distribution:** Like in the 1d-hybrid implementation, the radial points are split evenly between the $\Pi_r$ ranks in the radial loop. Additionally, the $N_\theta$ points are distributed evenly and contiguously between the $\Pi_\theta$ ranks (see Fig. 2a). At line 7, each rank is responsible for the computation of $N_\theta^- \times N_\phi \times N_r^-$ points. After the FFT at line 9 of Algorithm 1, $N_\theta^- \times N_m \times N_r^-$

points are stored locally, but the computation of the Legendre transform at line 10 requires that all $\theta$ angles for a given spherical harmonic order $m$ are available locally. This is guaranteed by the so-called $\theta$-transposition, which gathers the values for all $N_\theta$ $\theta$ angles while distributing the $m$-modes, resulting in $N_\theta \times N_m^- \times N_r^-$ local points.

The distribution of the $m$-modes is more involved than the distribution of the $\theta$-angles because there are $(N_\ell - m)$ different spherical harmonic degrees $\ell$ for each order $m$. Once again, we employ the "snake-like" distribution depicted in Fig. 1, which

guarantees that the largest imbalance between any two MPI ranks is $\Pi_\theta - 1$ tuples. See Fig. 2c–d for a graphical representation of this step.





Just like in the 1d-hybrid implementation, an $\ell$-transposition is required before the $\ell$-loop in order to gather locally all $N_r$ radial points. This once again involves a complex redistribution of data points. Because of Eq. (17), we would like to distribute only the $\ell$-modes while keeping all radial points and $m$-modes local. This would incur an unfeasible volume of data transfer.

We opted for a compromise that exploits a special regularity of the current data structure and leaves the $m$-modes local. Only the $\Pi_\theta$ ranks having a common $m$-mode then need to communicate in the radial direction. Figure 2d–e visualizes how the respective communication works. Only the magenta and red ranks need to share their radial information since they represent the same $m$-modes. They then also have the same number of $\ell$-modes and can thus split the number of modes. The snake-like distribution leads to a good balancing for both ranks. The resulting distribution is visualized in Fig. 2e.

The figure also illustrates a drawback of this distribution: the $m$-modes for one $l$-mode can be found on different ranks. This means that the $\mathbf{B}_\ell^i$ matrices have their columns distributed between MPI ranks and that the time-stepping matrices $\mathbf{M}_\ell^{\delta t}$ and their respective decomposition need to be *duplicated* in $\Pi_\theta$ ranks. However, our numerical experiments show that the communication cost far outweighs the cost associated with solving Eq. (17), thus justifying our choice. This will be further discussed in section 4. Finally, we would like to mention that a similar data layout was employed by Marti and Jackson (2016) in their two-dimensional MPI decomposition of their pseudo-spectral code in spherical geometry.

### 3.3   Radial Loop Implementation

The spherical harmonics transform (SHT) takes a substantial portion of the runtime. MagIC relies heavily on SHTns (Schaeffer, 2013), a highly optimized library dedicated to computing the spherical harmonics transform using OpenMP, advanced SIMD vectorization and cache-hitting strategies (Ishioka, 2018). SHTns has been developed to compute the steps synthesized in lines 8–10 and lines 4–6, that is, independent of the radial level $r$. It can not only handle scalar fields but is also optimized for general vector fields and vector fields with a poloidal-toroidal decomposition. SHTns is written in C but has a Fortran interface with a dedicated data-layout for MagIC. Although SHTns is flexible and modular, it does not have an MPI-distributed interface. However, it offers an interface for computing the Legendre transform (lines 5 and 10 of Algorithm 1) only for an individual $(m,r)$ pair, which is the use case in our 2d-MPI implementation. The transposition and FFT computations are then handled separately by MagIC.

Next we describe how both the 1d-hybrid and 2d-MPI version of the code use SHTns for computing the SHT.

**1d-hybrid radial loop:** The spherical harmonics transforms are delegated directly to SHTns (see the pseudocode in Algorithm 2). For instance, for the temperature scalar field on the numerical grid, $T(\theta,\phi,r_k)$, a single call to spat_to_SH returns the fully transformed $T_{\ell m}(r_k)$ for a given radial level $r_k$. This is only possible because all $(\theta,\phi)$ points are local in the 1d-hybrid version of the code. No explicit optimization or multithreading is required, as SHTns handles this internally. The number of vector transforms needed per radial loop depends on the set of equations. Dynamo models under the Boussinesq approximation of the Navier-Stokes equations considered here (Eq. (1)—(4)) involve four vector transforms and one scalar transform for the backward SHTs and three vector transforms for the forward SHTs.

**2d-MPI radial loop:** the pseudocode for our implementation is shown in Algorithm 3. Since shtns_ml is called for single modes only, the implementation cannot benefit from the efficient internal OpenMP parallelization of SHTns. The computation





---

**Algorithm 2** Pseudocode for the 1d-hybrid radial loop

---

1: **for** each $r$ in $N_r^-$ **do**

2:     **for** each field **do**

3:         **call** shtns() *! backward SHT, $(\ell, m) \rightarrow (\theta, \phi)$*

4:     **end for**

5:     compute non-linear terms

6:     **for** each field **do**

7:         **call** shtns() *! forward SHT, $(\theta, \phi) \rightarrow (\ell, m)$*

8:     **end for**

9:     compute misc terms

10: **end for**

---

**Algorithm 3** Pseudocode for the 2d-MPI radial loop

---

1: **for** each $r$ in $N_r^-$ **do**

2:     **for** each field $g_i$ **do**

3:         **for** each local $m$ **do**

4:             $h_i(\cdot, m, r) = \text{shtns\_ml}(g_i(\cdot, m, r))$ *! Legendre transform*

5:         **end for**

6:         **call** queue_transposition($h_i$, $f_i$)

7:     **end for**

8:     **call** finish_queue() *! $\theta$-transposition and fftw*

9:     compute non-linear terms

10:     ...

11: **end for**

---

time for only a single mode is too small to justify the initialization and termination of a multithreading context. In addition, the larger number of SHTns calls causes some overhead. Some speed-up could possibly be gained with an OpenMP-parallelization of the SHTns calls (lines 3–5 in Algorithm 3) or directly at the radial loop level (line 1 in Algorithm 3). Some preliminary tests showed promising results for the former. However, here we shall continue discussing only the pure MPI version.

The so-called $\theta$-transposition happens in line 8 of Algorithm 3 and involves MPI communication. Pointers to the input fields $h_i$ and output fields $f_i$ are ordered in a queue. Upon calling `finish_queue`, each $h_i$ is effectively transposed. After receiving the data, a complex-to-real FFT (e.g. using Intel MKL) is performed and saved into the output fields $f_i$. The queue has a limited size. Once the limit is reached, `finish_queue` is called, immediately triggering the data transfer and the FFT computation. Additionally, upon exiting the loop, `finish_queue` is called one last time to treat any remaining field in the queue.

The $\theta$-transposition is implemented using the following algorithms:





- **P2P:** each $\theta$-rank performs one call to `mpi_isend` and `mpi_irecv` for each other $\theta$-rank (i.e. $\Pi_\theta - 1$) per scalar field, followed by a `mpi_waitall` call. MPI types are used to stride the data when needed. The advantage of this algorithm is that it does not require any particular packing and/or reordering of the data.

- **A2AW:** a single call to `mpi_alltoallw` per scalar field, using the same MPI types used for the P2P algorithm.

- **A2AV:** a single call to `mpi_alltoallv` per scalar field. No MPI types are needed, but the data needs to be reordered in a buffer prior to the call.

Two parameters need to be determined, namely, which MPI algorithm to use and the length of the queue. The choice of the MPI algorithm depends on the MPI implementation, the hardware and the number of resources. We compare the performance of each variant in subsection 4.1.

As for the length of the $\theta$-transposition queue, several factors must be taken into account. Longer queues allow more scalar fields to be packed per call, thus allowing larger message size, decreasing the impact of the latencies of the MPI calls. On the other hand, longer queues require larger send and receive buffers. The benefit of having larger messages sizes may be not relevant for all configurations, especially if all $\theta$-ranks are located within the same NUMA domain (for instance, Intel implementation of `mpi_alltoallv` automatically uses shared memory operations in this regime). Furthermore, larger queue

sizes do not guarantee full usage of the buffers. We discuss this in more detail in subsection 4.1.

### 3.4   $\ell$-loop Implementation

As discussed in subsection 3.2, the data needs to be redistributed with the so-called $\ell$-transposition prior to the execution of this part of the code. The distribution of the $\ell$-modes and their respective $m$-modes to the MPI processes has already been discussed in subsection 3.2.

Most of the walltime of the $\ell$-loop is spent in the $\ell$-transposition and in the backward substitution for solving the linear systems from Eq. (17). Depending on how often $\delta t$ changes, the cost for computing the LU decomposition for each dense matrix $\mathbf{M}_\ell^{\delta t}$ might become a bottleneck, but in the scenario investigated during our benchmarks, this was not the case. Nevertheless, a frequent change in $\delta t$ indicates that the algorithm should be run with a generally smaller time step.

Next we describe the $\ell$-loop implementations of both MagIC versions.

**1d-hybrid $\ell$-loop:** The pseudocode is shown in Algorithm 4. The $\ell$-transposition itself takes place in lines 1 and 20 of Algorithm 4. Inside one MPI rank, the work is parallelized with OpenMP. We have a two-level loop structure, an outer loop over the different $\ell$ degrees on the MPI rank and for each $\ell$ a loop over all possible $m \leq \ell$. A single OpenMP parallelization of the outer loop would directly compete with the MPI parallelization and would also lead to a strong load imbalance as the different harmonic degrees have different numbers of $m$. To avoid nested OpenMP regions (for outer and inner loop) we use instead

the OpenMP tasking approach to balance the work and to combine the parallelism of the outer and inner loop. We create an OpenMP task for each outer loop iteration which contains the LU decomposition of the matrix and a task creation loop for the inner $m$-loop. To minimize the overhead for OpenMP task creation, we use chunks of several $m$ together in one task.





---

**Algorithm 4** Pseudocode for the 1d-hybrid $\ell$-loop

---

1: $\ell$-transposition: $N_{\ell m} \times N_r^- \to N_{\ell m}^- \times N_r$

2: OMP PARALLEL

3: OMP SINGLE

4: **for** each local $\ell$ **do**

5:     OMP TASK

6:     **if** $\delta t$ changed **then** recompute: $\mathbf{M}_\ell^{\delta t} \in \mathbb{R}^{N_r \times N_r}$ and its factors

7:     **for** all chunks of $m \in \{0, \ell\}$ **do**

8:         OMP TASK

9:         **for** all $m$ in chunk **do**

10:             update $\boldsymbol{B}_{\ell m}^i$

11:             solve $\mathbf{M}_\ell^{\delta t} \boldsymbol{G}_{\ell m}^{i+1} = \boldsymbol{B}_{\ell m}^i$

12:         **end for**

13:         OMP END TASK

14:     **end for**

15:     update radial derivatives

16:     OMP END TASK

17: **end for**

18: OMP END SINGLE

19: OMP END PARALLEL

20: $\ell$-transposition: $N_{\ell m}^- \times N_r \to N_{\ell m} \times N_r^-$

---

---

**Algorithm 5** Pseudocode for the 2d-MPI $\ell$-loop

---

1: $\ell$-transposition: $N_{\ell m} \times N_r^- \to N_{\ell m}^- \times N_r$

2: **for** each local $\ell$ **do**

3:     **for** each local $m \leq \ell$, update $\boldsymbol{B}_{\ell m}^i$

4:     **if** $\delta t$ changed **then** recompute: $\mathbf{M}_\ell^{\delta t} \in \mathbb{R}^{N_r \times N_r}$ and its factors

5:     **for** each local $m \leq \ell$, solve $\mathbf{M}_\ell^{\delta t} \boldsymbol{G}_{\ell m}^{i+1} = \boldsymbol{B}_{\ell m}^i$

6:     update radial derivatives

7: **end for**

8: $\ell$-transposition: $N_{\ell m}^- \times N_r \to N_{\ell m} \times N_r^-$

---





**2d-MPI $\ell$-loop:** The pseudocode for the $\ell$-loop in the sequential and the 2d-MPI implementations (see Algorithm 5) differs only in the $\ell$-transpositions and the number of $\ell$- and $m$-modes stored locally (which consequently affects the amount of linear
systems and right hand sides in each rank). The distribution of the $(\ell, m)$ tuples has already been discussed in subsection 3.2. While the difference in the number of modes has an impact on the computation of the solution of Eq. (17), this pales in comparison with the time spent in communication.

In both the 1d-hybrid and 2d-MPI implementations the $\ell$-transposition might be performed using the P2P, A2AV and A2AW algorithms discussed previously. Additionally, the following algorithm has been implemented:

– **A2AP:** a single call to `mpi_alltoall` per scalar field with padding to accommodate the largest message. No MPI types are needed, but the data needs to be reordered in a buffer. Due to the intrinsic imbalance in the distribution of the data, considerably larger buffers are required. Some MPI libraries implement faster algorithms for `mpi_alltoall`, which can in some cases be advantageous.

In the 1d-hybrid implementation of MagIC an auto-tuning routine always selects the optimal transposition strategy during the
initialization stage of the code. For the 2d-MPI version, the algorithm still must be selected manually via the parameter file. We discuss the performance of the different strategies in subsection 4.4.

## 4 Benchmark

In this section we compare the performance of the 1d-hybrid and 2d-MPI implementations in a practical, realistic setting. We also profile some crucial sections of the code in order to highlight the advantages and shortcomings of the different implemen-
tations.

Following Matsui et al. (2016), we adopt the dynamo benchmark "BM1" from Christensen et al. (2001) to assess the performance of the code. The benchmark uses a Rayleigh number of $10^5$, a magnetic Prandtl number of five and a Prandl number of one. Both boundaries are assumed to be rigid, electrically non-conducting and held at a fixed temperature. For each radial point there are 13 scalar fields (four vector fields and one scalar field) to be $\theta$-transposed during the backward SHT and nine
scalar fields (three vector fields) during the forward SHT. For all tests, we set $N_\phi = 2N_\theta$ and $N_\ell = 2N_\theta/3$, meaning that the geometry is completely described by $N_\theta$ and $N_r$.

All tests were performed on the Cobra cluster at the Max Planck Computing and Data Facility (MPCDF). Each node of this machine possesses two Intel Xeon Gold 6148 processors, and 192GB of main memory. Each processor has a single NUMA domain with 20 cores. The nodes are interconnected with a 100 Gb/s Omnipath interconnect and a non-blocking fat-tree
topology. The workload is managed by Slurm, and hyperthreading is always disabled. The pinning is done automatically by Slurm and packs all $\theta$-ranks within a single NUMA domain whenever $1 < \Pi_\theta \leq 20$. We compiled all software using Intel Fortran Compiler 2021.2.0 with Intel MPI 2021.2.0 and Intel MKL 2020.4. The kernels of SHTns were compiled using GCC 10.2.0 for higher performance.





The runtime of each crucial section was measured for each MPI rank independently using perflib, a lightweight profiling
library developed in-house. Our figures show the average over all ranks as a solid line. A coloured background shows the
spread between the fastest and slowest ranks and visualizes the imbalance of the times measured.

### 4.1   $\theta$-transpose benchmark

The performance of this code section is of special interest. This is because the $\theta$-transposition is required only for the 2d-MPI
implementation, thus adding unavoidable extra cost for handling the two-dimensional distribution of the data.

For the tests we fixed the size of a scalar field in grid space to 240 KiB per rank per radial level. This can be achieved by
using the formula $N_\theta = 48\sqrt{10\Pi_\theta}$ to determine the number of latitudinal points.

We expect the performance of the $\theta$-transposition to strongly depend on the distribution of the $\theta$-ranks across the computing
nodes. To explore this behaviour, we defined three communication regimes with the following parameters:

- **intranuma**: $\theta$-communication is constrained within a single NUMA domain: $\Pi_\theta = 10, \Pi_r = 4$, $N_\theta = 480$ and $N_r = 128$.

- **intranode**: $\theta$-communication is constrained within a single node, but between NUMA domains: $\Pi_\theta = 40, \Pi_r = 1$, $N_\theta = 960$ and $N_r = 32$.

- **internode**: $\theta$-points are communicated between nodes. We set $\Pi_\theta = 40, \Pi_r = 2$, $N_\theta = 960$ and $N_r = 64$ and we force half $\theta$-ranks to communicate across two nodes by pinning the MPI ranks with `-cpu-bind=map_cpu` option.

We measure the performance of the $\theta$-transposition in these three different regimes and determine for which algorithm
(A2AV, P2P or A2AW, discussed in section 3) and for which queue length (for Algorithm 3) they perform best. Notice that
larger queue lengths do not *guarantee* full usage of the larger buffers. As an example, suppose that $q_{max}$ is the queue length and
that $q$ fields are queued for transposition. If three new fields are added to the queue (i.e. a vector field), but the $q + 3 > q_{max}$,
the algorithm transposes the first $q$ fields, clears the queue and then adds the three fields into a new queue. This could be
improved to enforce full buffer usage in future versions, but our benchmarks demonstrate that the benefits would be marginal.
Table 1 shows the effective usage of the transposition buffers for queue lengths from 3 to 13, as well as the "queue-free" version
denoted by queue length one.

Figure 3 shows the times for 50 time steps for the three different communication regimes. The inferior performance of the
A2AV implementation is attributed to the time spent reordering the send and receive buffers. Both A2AW and P2P use the
same MPI types to avoid this reordering and therefore perform very similar. Concerning the queue length, the intranuma and
intranode regimes perform best with smaller queues, while the opposite is true for the internodes regime.

Since the A2AW implementation for the $\theta$-transposition seems to perform best, we stick to this scheme for the following
test where we address the impact of the queue length.



**Table 1.** Effective usage of $\theta$-transposition buffers (for a single radial point and time step) for the A2AW algorithm. Each number represents how many scalar fields were packed in an individual call to `mpi_alltoallw`, for a single time step and radial point. The average message size was computed assuming 240KiB per scalar field in each rank.

| queue length | $\theta$-transpose backward | forward | avg. usage | avg. message size |
|---|---|---|---|---|
| 1 | 13x1 | 9x1 | 1.0 | 240KiB |
| 3 | 1,3,3,3,3 | 3,3,3 | 2.8 | 660KiB |
| 4 | 4,3,3,3 | 3,3,3 | 3.1 | 754KiB |
| 5 | 4,3,3,3 | 3,3,3 | 3.1 | 754KiB |
| 6 | 4,6,3 | 6,3 | 4.4 | 1056KiB |
| 7 | 7,6 | 6,3 | 5.5 | 1320KiB |
| 8 | 7,6 | 6,3 | 5.5 | 1320KiB |
| 9 | 7,6 | 9 | 7.3 | 1760KiB |
| 10 | 10,3 | 9 | 7.3 | 1760KiB |
| 11 | 10,3 | 9 | 7.3 | 1760KiB |
| 12 | 10,3 | 9 | 7.3 | 1760KiB |
| 13 | 13 | 9 | 11.0 | 2640KiB |

## 4.2 2d-MPI SHT Benchmark

In this subsection we continue the comparison between intranuma, internuma and internode regimes, but now bring the whole
spherical harmonics transform into perspective. Since this constitutes one of the largest portions of the computation, it is particularly interesting to determine the behaviour in the different communication regimes.

Figure 4 shows the $\theta$-transposition time as well as the computation time spent inside the FFT and Legendre transform calls. Although the computation time itself should not depend on the queue length, Figure 4 shows a small but consistent influence. We assume that this is linked to the change in data access pattern caused by queuing and buffering data for the $\theta$-transposition.

Even for the intranuma case shown in Fig. 4a, where a queue length of three provides a small performance boost for the $\theta$-transposition, the performance of the SHT is optimal for queue length of one. In the following test we therefore fix the queue length to one for intranuma and intranode regimes and to 13 for the internode regime.

## 4.3 $\ell$-transpose strong scaling benchmark

In this subsection we compare the performance of the four different $\ell$-transposition implementations for the 2d-MPI version.
We would like to recall that the 1d-hybrid implementation uses an auto-tuning routine which automatically selects the best algorithm during the initialization. Moreover, the main difference between 1d-hybrid and 2d-MPI implementations in the $\ell$-



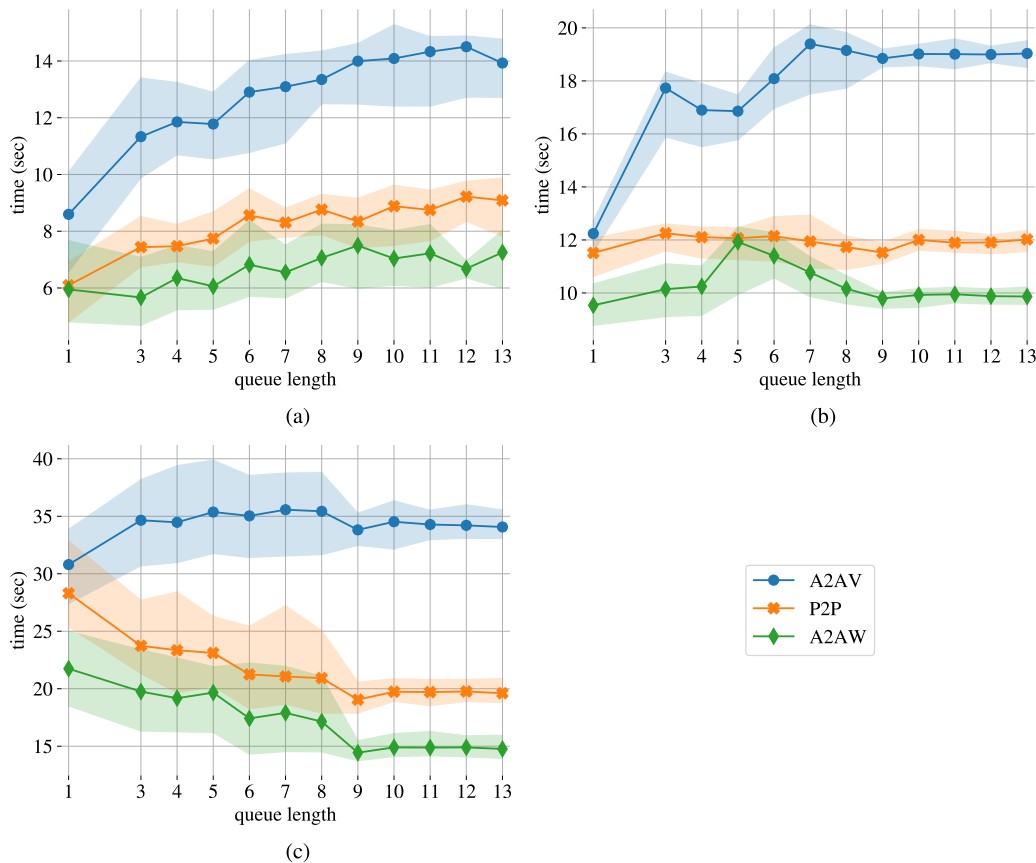

**Figure 3.** Runtime for the $\theta$-transpose in the intranuma (a), intranode (b) and internode (c) regimes for varying queue length and 50 time steps.

loop is the distribution of the $(\ell, m)$ tuples, as already discussed in subsection 3.2. The communication pattern therefore differs significantly for the two implementations.

The problem we chose for this and subsequent subsections is the dynamo benchmark with $N_r = 301$ and $N_\theta = 960$. We
compute 100 time steps for several values of $\Pi_r$ and $\Pi_\theta$, and for core counts ranging from 120 to 24,000 (3 to 600 nodes, respectively). For the 1d-hybrid implementation, we use 10 OpenMP threads per MPI task for tests with up to 1,000 cores (25 nodes) and 20 threads for larger runs (this configuration resulted in the best total time). The 1d-hybrid version of the code can use at most 6,000 cores (150 nodes) with one radial point per socket ($\Pi_r = 300$ and 20 threads per MPI task).

For the 2d-MPI scheme, we fix the $\theta$-transposition implementation to A2AW (as discussed in subsection 4.1) and the queue
lengths to one for $\Pi_\theta = 10, 20$, or 40 and 13 otherwise. This new version of MagIC admits more combinations of $\Pi_\theta$ and $\Pi_r$ and allows the use 24,000 cores (600 nodes). For some core counts, multiple values of $\Pi_\theta$ and $\Pi_r$ were allowed. For example, for 6,000 cores one may choose $\Pi_\theta = 20$ or $\Pi_\theta = 40$ with $\Pi_r = 300$ or $\Pi_r = 150$. We tested both and kept the configuration



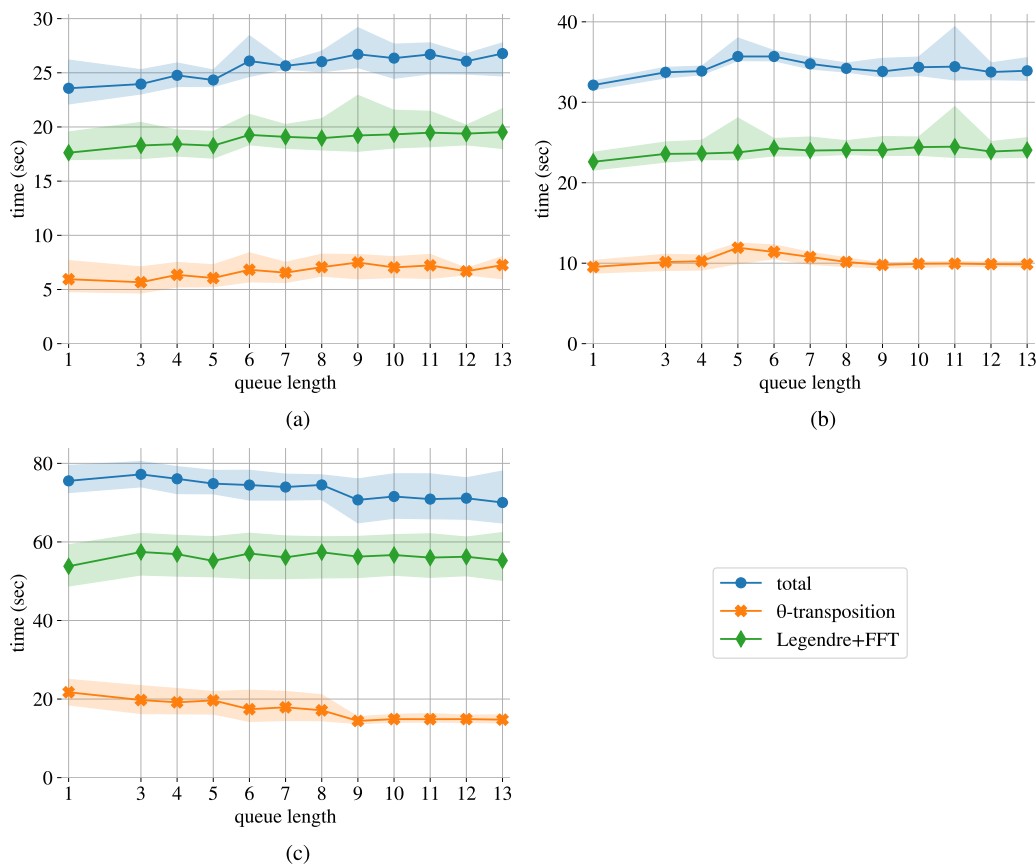

**Figure 4.** Runtime for the different components of the SHT in the intranuma (a), intranode (b) and internode (c) regimes. Data for 50 time steps using the A2AW implementation.

with the best total time. For the reader's convenience, all plots show the value of $\Pi_\theta$ for the 2d-MPI version at the top horizontal axis. We also set the experiments such that $N_r - 1$ is divisible by $\Pi_r$, and assign the last radial point to the last radial MPI

rank. This is currently a limitation of MagIC. In principle, this should cause imbalance in some cases e.g. with $\Pi_r = 300$ the last radial MPI rank will store and solve twice the number of radial levels as the other ranks. However, the last radial MPI rank receives less $(\ell, m)$ tuples, which diminishes the impact of the imbalance. Since both codes suffer from the same imbalance, we opt for simplifying the analysing to the average runtime only.

In Fig. 5 we can see that, except for the A2AP variant, the 2d-MPI implementation is always significantly faster than the

1d-hybrid implementation. This is a direct consequence of the communication-avoiding pattern of the 2d-MPI distribution.

The 1d-hybrid version shows a good strong scalability, which, however, starts to degrade after 1,000 cores (25 nodes). At this core count the 1d-hybrid version transitions from 10 to 20 threads. OpenMP efficiency is typically difficult to maintain for large thread counts and this is likely the reason for the performance degradation.



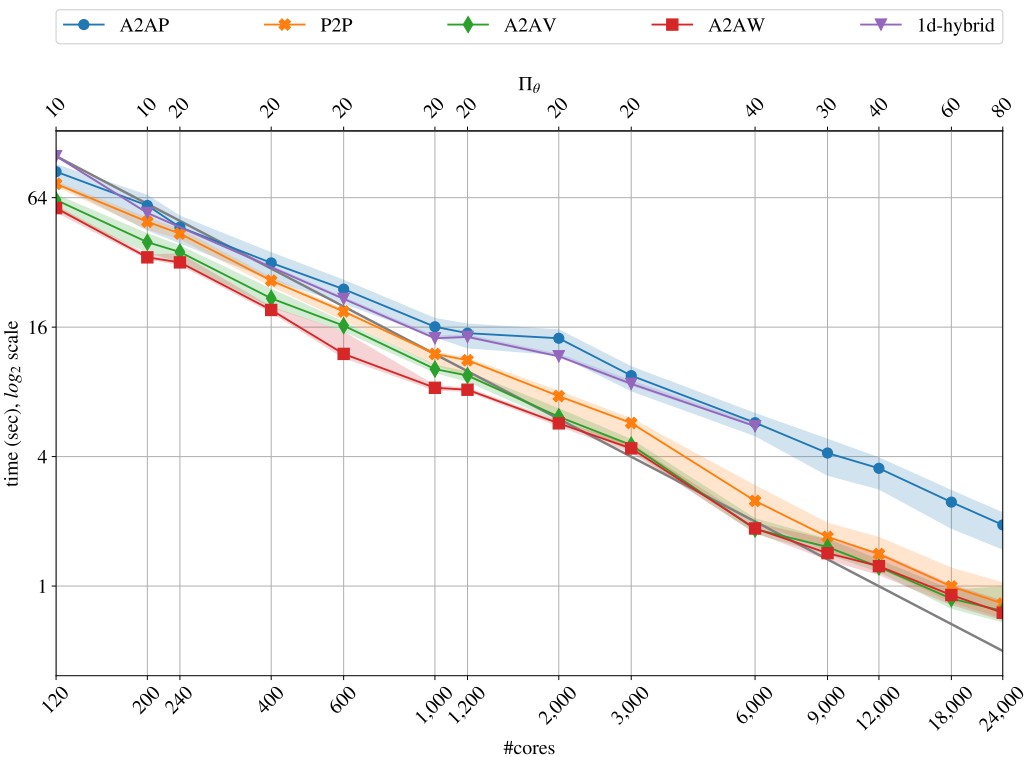

**Figure 5.** Strong scalability of $\ell$-transposition for 100 time steps of the dynamo benchmark ($N_r = 301$ and $N_\theta = 960$).

The 2d-MPI implementation, especially for the A2AW variant, scales remarkably well up to 6,000 cores (150 nodes), where the performance starts to degrade. However, the losses remain acceptable up to the largest number of 24,000 cores we could test. The A2AV variant performs similarly to A2AW, and beyond 12,000 cores all variants show nearly identical timings. This test suggests to prefer A2AW which will be kept in the following. However, as the behaviour may be different on other architectures and for other MPI libraries, an auto-tuning routine that selects the fastest option in the initialization phase of a simulation seems a good idea for the future.

## 4.4 $\ell$-loop strong scaling

In this subsection we take a deeper look into the different parts of the $\ell$-loop. We are especially interested in the impact of the 2d-MPI data distribution on the LU decomposition and on the backward substitution. For these tests we use the same setting as in the previous subsection (100 time steps, $N_r = 301$ and $N_\theta = 960$) and stick to the A2AW implementation of the transpositions.



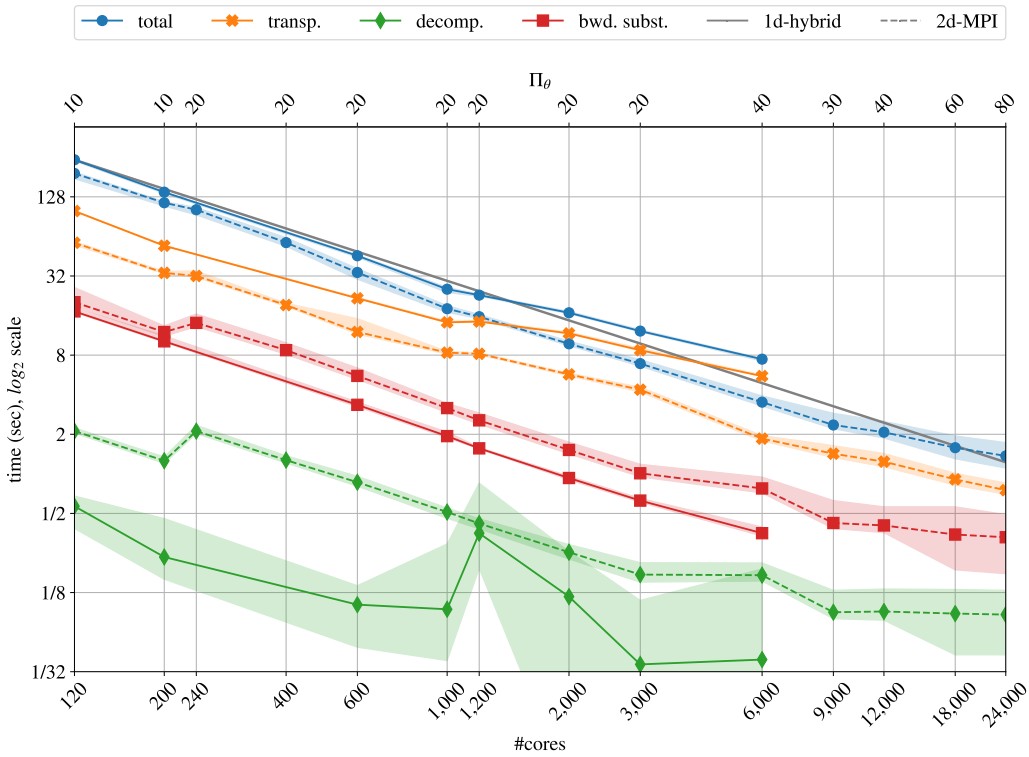

**Figure 6.** Strong scalability for the $\ell$-transposition, LU decomposition, and backward substitution for the dynamo benchmark.

In Fig. 6 we compare transposition time, LU decomposition time, and the backward substitution time (referring to Eq. (19)) for the 1d-hybrid and the 2d-MPI implementations. In addition, the total times for the $\ell$-loop are shown, which also include the updating of time derivatives and related operations.

In both the 1d-hybrid and 2d-MPI implementations, the performance is dominated by the transposition time. Here the 2d-MPI clearly outperforms the 1d-hybrid implementation because of the reduced communication, as discussed in subsection 3.2.

The timing results in Fig. 6 demonstrate that decomposition and backward substitution for the linear systems do not scale well with $\Pi_\theta$ (upper horizontal axis). As explained in subsection 3.2, the data are distributed in the direction of the $m$-modes, but the time-stepping matrices, which need to be LU-decomposed, are independent of $m$.

For a fixed $\Pi_r$, the decomposition time is thus expected to remain independent of $\Pi_\theta$. Likewise, the backward substitution time should just mildly be impacted by larger $\Pi_\theta$. This can be clearly seen in Fig. 6: 120 and 240 cores have $\Pi_r = 12$, with $\Pi_\theta = 10$ and 20 respectively. Their decomposition time is roughly the same, with a slightly faster backward substitution time for 240 cores. The same can be observed for 200 and 400 cores as well as 3,000 and 6,000 cores. There is a spike in the decomposition time for the 1d-hybrid implementation at 1,200 cores.





However, the performance gain due to the communication-avoiding distribution of the 2d-MPI implementation far outweighs these shortcomings. We would like to highlight that it is only possible to reduce the communication volume due to the particular

distribution pattern shown in Figure 2, i.e. each $\theta$-rank communicates with $\Pi_r$ ranks in the radial direction. The communication-avoiding scheme discussed here is not possible for the 1d-hybrid implementation, since one of these directions is missing. This specific feature of the 2d-MPI version is essential to enable a better overall performance for large cores count, as we shall discuss in the next subsection.

## 4.5 Overall performance and strong scaling comparison

Finally we discuss the strong scaling of the MagIC time step, i.e. of the main application without initialization, finalization and diagnostics. As usual, we define the parallel efficiency as

$$\epsilon = \frac{t_{\text{ref.}} \times n_{\text{ref.}}}{t \times n} \tag{20}$$

where $t$ is the main application time in seconds and $n$ is the number of cores used. We use the test with $n_{\text{ref.}} = 120$ as a reference.

Following Matsui et al. (2016), we consider that $\epsilon > 0.6$ is a sufficiently good efficiency. Additionally, we define the "parallel cross-efficiency" $\epsilon_*$ that compares the 2d-MPI timing to the 1d-hybrid reference time $t_{\text{ref.}} = 405.6\,\text{s}$ for $n_{\text{ref.}} = 120$ as a baseline. The timings and parallel efficiency, along with other information about the tests, are listed in Table 2. The graphical visualization of these values (discriminating also the time for $\ell$-loop and radial loop) is shown in Fig. 7. Since we use the same problem size and settings already discussed in subsection 4.3 and subsection 4.4 the curves for the $\ell$-loop are the same as in

Fig. 6.

In Fig. 7 it is interesting to notice that the performance of the 1d-hybrid implementation is dominated by the $\ell$-loop performance, whereas the performance of the 2d-MPI implementation is dominated by the performance of the radial loop. To some degree this is expected, because the radial loop of the 2d-MPI implementation requires the $\theta$-transposition in addition to the SHT.

The radial loop for the 1d-hybrid code scales well up to 6,000 cores, but the inferior scalability of the $\ell$-loop already leads to a noticeable degradation beyond 1,200 cores in Fig. 7. This is the cause of the decrease in parallel efficiency $\epsilon$ apparent in Table 2 for 2,000 and more cores.

The 2d-MPI implementation starts suffering from a small loss of scalability of the $\ell$-loop at 9,000 cores but remains at an acceptable level up to 24,000 cores. The radial loop scales remarkably well up to 12,000 cores but shows degraded performance

at 18,000 and 24,000 cores. This can be explained by the problem size being too small for such numbers of cores, which is reflected by $N_r^- = 1$ and $N_\ell^- \approx 22$ for 9,000 cores. Both the $\ell$ and the radial loop contribute to the loss of overall performance at the highest core counts apparent in Fig. 7.

The parallel cross-efficiency (see Table 2 and Fig. 7) shows that the new 2d-MPI version is about 20% slower than the 1d-hybrid implementation for core counts of $n < 2000$. However, the 2d-MPI variant scales better for $n \geq 2,000$ with $\epsilon = 0.7$





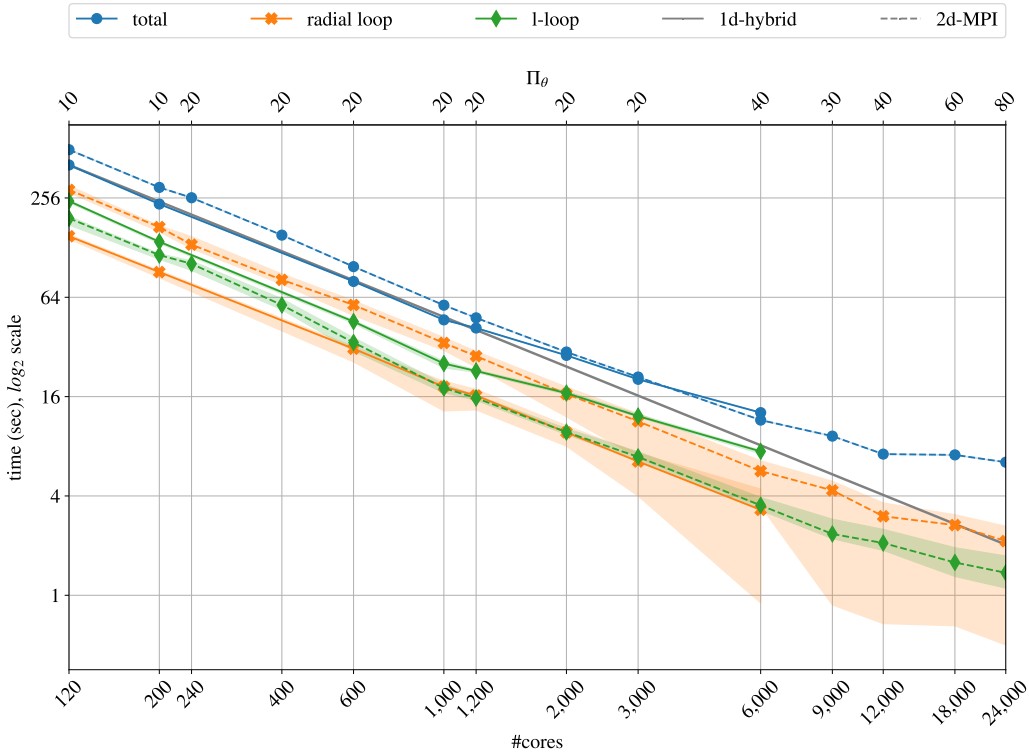

**Figure 7.** Strong scalability for the 1d-hybrid and 2d-MPI implementations of MagIC for 100 time steps of the dynamo benchmark using $N_r = 301$ and $N_\theta = 960$.

for up to $n = 12,000$. The 1d-hybrid implementation, on the other hand, reaches $\epsilon = 0.63$ for 6,000 cores. The 2d-MPI is as fast as the 1d-hybrid implementation at $n = 3,000$ but already 10% faster at $n = 6,000$.

     The benefits offered by the 2d-MPI implementation can be illustrated with a simple example. Instead of performing a simulation with the 1d-hybrid implementation on 6,000 cores a scientist could opt for using the 2d-MPI implementation on 12,000 cores and obtain the solution in only 56% of the runtime. In other words, by investing only 12.5% more CPU-hours,

the 2d-MPI implementation arrives at the same solution in half the time.

     The parallel cross-efficiency also gives an idea on how "far" the parallel efficiency of the 2d-MPI is from the 1d-hybrid implementation. There is a significant gap for 120 to 1,200 cores, but the gap closes at about 2,000 cores, e.g. $\epsilon = 0.86$ for the 1d-hybrid and $\epsilon_* = 0.82$. This trend continues until 6,000 cores, when the parallel cross-efficiency overtakes the parallel efficiency of the 1d-hybrid implementation.





**Table 2.** Parallel efficiency and cross-efficiency the 1d-hybrid and 2d-MPI implementations, detailing $\Pi_r$ and $\Pi_\theta$ and the main runtime in seconds.

| #cores | 1d-hybrid | | | 2d-MPI | | | |
|---|---|---|---|---|---|---|---|
| | $(\mathbf{\Pi_r}, \mathbf{\Pi_\theta})$ | time | $\epsilon$ | $(\mathbf{\Pi_r}, \mathbf{\Pi_\theta})$ | time | $\epsilon$ | $\epsilon_*$ |
| 120 | (12, 10) | 405.6 | 1.00 | (12, 10) | 502.1 | 1.00 | 0.81 |
| 200 | (20, 10) | 235.1 | 1.03 | (20, 10) | 296.6 | 1.02 | 0.82 |
| 240 | - | - | - | (12, 20) | 256.8 | 0.98 | 0.79 |
| 400 | - | - | - | (20, 20) | 152.4 | 0.99 | 0.80 |
| 600 | (60, 10) | 79.8 | 1.02 | (30, 20) | 98.2 | 1.02 | 0.83 |
| 1,000 | (100, 10) | 46.7 | 1.04 | (50, 20) | 57.3 | 1.05 | 0.85 |
| 1,200 | (60, 20) | 41.7 | 0.97 | (60, 20) | 48.0 | 1.05 | 0.84 |
| 2,000 | (100, 20) | 28.4 | 0.86 | (100, 20) | 29.8 | 1.01 | **0.82** |
| 3,000 | (150, 20) | 20.4 | 0.80 | (150, 20) | 21.0 | 0.95 | **0.77** |
| 6,000 | (300, 20) | 12.8 | **0.63** | (150, 40) | 11.5 | 0.87 | **0.70** |
| 9,000 | - | - | - | (300, 30) | 9.2 | 0.73 | 0.59 |
| 12,000 | - | - | - | (300, 40) | 7.2 | 0.70 | 0.57 |
| 18,000 | - | - | - | (300, 60) | 7.1 | 0.47 | 0.38 |
| 24,000 | - | - | - | (300, 80) | 6.4 | 0.39 | 0.32 |

## 4.6 Analysis of the radial loop

The parallel cross-efficiency in Table 2 shows that the performance of both implementations is rather close for core counts larger than 2,000. This rightfully raises the question if this small gap could be bridged. Furthermore, it has already been established in previous sections that the main bottleneck of the 2d-MPI implementation is the radial loop, but the $\theta$-transposition and the computation time (i.e. mostly the SHT) of the radial loop have not yet been thoroughly compared. The goal of this subsection is to provide this comparison, discuss which portions of the code could be optimized, and what impact this could have on the 2d-MPI implementation.

For the 1d-hybrid code, let $t_t$ and $t_r$ respectively denote the full time of the main application and the radial loop only, both given in seconds. Analogously, for the 2d-MPI code, let $T_t$ denote the time of the full main application and $T_r + T_\theta$ denote the time of the radial loop only, where $T_r$ is the computation time only and $T_\theta$ is the $\theta$-transposition time only (including communication and necessary buffer copies, if any). Both $t_t$ and $T_t$ are shown in Table 2. Table 3 shows $t_r$, $T_r$ and $T_\theta$ for the same experiment, from 120 to 6,000 cores.

Table 3 shows that the $\theta$-transposition times comprises on average 17.5% of the radial loop (except for 6,000 cores, where it takes circa 28% of time of the radial loop). The runtime of $\theta$-transposition can be further optimized e.g. by introducing complex strategies such as communication-computation overlapping. However, from Table 3 it is evident that the main bottleneck is indeed the computation time $T_r$.





**Table 3.** Radial loop performance comparison for the dynamo benchmark. $t_r$ (resp. $T_r$) is the *computation* time for the radial loop of the 1d-hybrid (resp. 2d-MPI) implementation and $T_\theta$ is the time for the $\theta$-transposition in the 2d-MPI implementation. All times are given in seconds. The columns $\rho$ and $\kappa$ are defined in Eq. (21) and Eq. (22) respectively.

| #cores | $t_r$ | $T_r$ | $T_\theta$ | $\rho$ | $\kappa$ |
|---|---|---|---|---|---|
| 120 | 150.1 | 240.7 | 45.7 | 1.60 | 0.60 |
| 200 | 91.0 | 144.8 | 25.8 | 1.59 | 0.58 |
| 600 | 31.2 | 45.3 | 12.2 | 1.45 | 0.60 |
| 1,000 | 18.4 | 27.0 | 6.8 | 1.47 | 0.61 |
| 1,200 | 16.3 | 22.6 | 5.5 | 1.39 | 0.72 |
| 2,000 | 9.7 | 13.3 | 3.3 | 1.37 | 0.90 |
| 3,000 | 6.5 | 9.0 | 2.3 | 1.39 | 0.93 |
| 6,000 | 3.3 | 4.0 | 1.6 | 1.23 | 1.31 |

The next natural question is how the computation time of the radial loop of both implementations compare with each other. For that we use the ratio

$$\rho = \frac{T_r}{t_r}. \tag{21}$$

Ideally, $\rho$ would be always close to one, in which case the computational efficiency of the radial loop of both implementations would be identical. In practice, distributing the data between more MPI ranks means that each rank has less local data, which makes cache-hitting, vectorization, and other fine-grained optimization techniques less efficient. In other words, even a finely optimized implementation of the 2d-MPI data distribution algorithm is likely to have $\rho > 1$.

The column $\rho$ from Table 3 shows that the performance gap is far from one in most cases and is inversely proportional to the number of cores. This means that bridging the large gap in performance for the lower core counts might be unfeasible, but for larger core counts it is attainable.

We now attempt to determine how much improvement in the 2d-MPI implementation is required for its main application runtime (including all transposition times) to match the runtime of the 1d-hybrid code. In practice, several portions of the 2d-MPI code could benefit from cache-hitting strategies, fine tuned vectorization, computation-communication overlapping, amongst others. We will now discuss a much simpler scenario in which *only* the computation time of the radial loop can be improved, from $T_r$ to $\kappa T_r$. Subtracting $T_r$ from the total time $T_t$ and adding the "new" runtime $\kappa T_r$, we obtain the following equation

$$T_t - T_r + \kappa T_r = t_t, \quad \kappa = \frac{t_t - T_t}{T_r} + 1. \tag{22}$$

which gives us the needed improvement factor $\kappa$. The values are shown in Table 3.

Table 3 shows that, for low core counts, improvements in $T_r$ alone could not bridge the gap in the performance between 2d-MPI and the 1d-hybrid implementations. From 120 to 1,000 cores we have $t_r > \kappa T_r$, meaning that the 2d-MPI implementation





would have to compute the radial loop *faster* than the already highly optimized 1d-hybrid code for both codes to perform the same.

The last three entries in Table 3 show a much more realistic improvement factor. An improvement of 10% (resp. 7%) in $T_r$ would suffice to bridge the gap between the main application time for 2,000 cores (resp. 3,000 cores). For 6,000 cores, Table 3 shows that $T_r$ could be 31% *slower* and both codes would still be equivalent. This is due to the poor scalability of the $\ell$-transposition of the hybrid code, which gives the 2d-MPI some "room" for performance loss in other portions of the code.

## 5   Conclusions and Future Work

We described a new parallelization scheme based on a two-dimensional, MPI-based data decomposition for MagIC, an open-source code for three-dimensional fluid dynamics simulations in spherical geometry, with high scientific impact in a broad range of scientific fields ranging from fundamental fluid dynamics and modeling of planetary dynamos to stellar dynamics. MagIC uses spherical surface harmonics of degree $\ell$ and order $m$ for the spectral representation in longitude and latitude and Chebyshev polynomials in radius. Our newly implemented 2d-MPI scheme is compared to the previously established 1d-hybrid code version that uses MPI and OpenMP and has been highly optimized over years.

Thanks to a number of new concepts, the 2d-MPI version presented here can already compete with the hybrid implementation in terms of runtime, and in addition offers the possibility to use a significantly larger number of cores. It opens the possibility to employ tens of thousands of CPU cores on modern HPC clusters and paves the way to using the next-generation CPU architectures.

Decisive for its success is a communication-avoiding data distribution of the $\ell$- and $m$-modes for the computation of the actual time step in MagIC and the optimization of the associated communications strategies. Compared to the existing one-dimensional data decomposition, the new two-dimensional data distribution scheme requires an additional, costly transposition in the azimuthal direction. This leads to a performance penalty at low core counts which might be hard to overcome. However, for the dynamo benchmark and the setups considered here, and starting at moderate core counts of two thousand, the performance of the new version is mostly limited by the not yet fully optimized main radial loop, which can be as much as 60% slower than its counterpart in the one-dimensional data distribution.

Our results showed that a mere 10% performance gain in the computation time of the radial loop of the 2d-MPI implementation would bridge the gap between both code versions for two thousand cores or more, in addition to providing an extended strong scalability regime. Other sets of equations solved by MagIC, such as the anelastic equations (Jones et al., 2011), may present a different cost associated with the radial loop and would require a separate investigation.

Our implementation paves the way towards a future, unified hybrid variant of MagIC which combines a two dimensional data distribution with the proven benefits of an additional OpenMP layer in order to further improve the computational performance across an even wider range of simulation scenarios. This future *"2d-hybrid"* version could retain the benefits of our communication-avoiding data distribution and its improved strong scaling behaviour, while still benefiting from the performance of the finely optimized multithreaded libraries used within the radial loop. We expect that the performance of the radial





loop of such a 2d-hybrid implementation will be closer to the performance of the 1d-hybrid version, effectively eliminating the
main bottleneck of the 2d-MPI version.

*Code availability.* MagIC's source code is under active development and availabe for download from https://github.com/magic-sph/magic
under the GNU GPL v3 license. The 1d-hybrid version discussed here is the commit c63930b of the master branch (version 5.10). A frozen
version is archived on Zenodo (https://doi.org/10.5281/zenodo.5204425). The 2d-MPI version is the commit 6cabfa7 of the merge-paral
branch on GitHub, branched from commit c63930b. A frozen version has also been archieved on Zenodo (https://doi.org/10.5281/zenodo.
595   5171780).

*Author contributions.* Gastine and Wicht developed and maintained the main branch of MagIC. Gastine also managed the use of external
libraries (FFTW, LAPACK, SHTns, etc) and fine optimization of OpenMP. Dannert conceptualized, implemented and tested the algorithms
for the 1d-hybrid implementation. Lago conceptualized, implemented and tested the algorithms for the 2d-MPI implementation. The nu-
merical experiments were designed and conducted by Lago with the supervision of Dannert and Gastine. Dannert and Rampp supervised
HPC aspects of the project. Wicht and Gastine were responsible for supervising and interpreting the geophysical aspects of the project. The
original draft was written by Lago. The reviewing and editing of the current draft was conducted equally by all authors.

*Competing interests.* The authors declare no competing interests.





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
