# Peer review of "MagIC v5.10: a two-dimensional MPI distribution for pseudo-spectral magnetohydrodynamics simulations in spherical geometry"

_Geoscientific Model Development, 2021_

## Author Response (AR1)

**Author's Response**

October 20, 2021

Dear Referee#1 and Referee#2,

We'd like to thank you for your very thorough and constructive review and for the suggestions and corrections which were proposed. In the following we individually address all points raised:

**Referee#1, Comment 1:** *First, I would like to point out that the present scaling tests are not in the practical range for the productive runs. Dynamo simulations in a rotating spherical shell are performed a few million time steps or even tens thousand steps. The minimum elapsed time is still more than one second in Figure 7, so it suggests that the present model needs approximately 12 days for one million steps. I guess that the practical problem size would be the half of the horizontal resolution for the productive runs. I recommend that the author describe the reason how they choose the spatial resolution and target elapsed time for the productive runs.*

The value of $\delta t$ (and thus, the total number of time steps) depends not only on the grid resolution, but also on the nature of the physical problem and its control parameters (Ekman, Prandtl and Rayleigh numbers). As an example, convection in thin spherical shells will require much larger spatial resolution but will converge in a significantly smaller number of iterations, than, say, geodynamo models geared to model reversals of the Earth magnetic field. As such, the total number of iterations is not a really meaningful measure.

However, as explained in lines 435–436 of the revised text, the times are measured for 100 timesteps. In the largest run (24,000 cores), the main application time was 6.4 seconds, thus one million timesteps would require 18 hours in average. For the the largest "recommended" run (12,000 cores, where the parallel efficiency remains within acceptable levels), one million time steps are estimated to require 20 hours in average.

Finally, the development of the 2d-MPI version is justified mostly for large grids. The chosen resolution is comparable to current state-of-the-art geodynamo models (e.g. Schaeffer et al. 2017). For smaller grids, the 1d-hybrid version of the code should deliver the optimal performance.

**Referee#1, Comment 2:** *Another question is that the authors perform data communications for each radial layer and each scalar component in the 2D parallelization. Calypso and Rayleigh perform these communications with single MPI_ISEND/IRECV or MPI_ALLREDUCEV, respectively. Can authors discuss the advantage of the present communications from Calypso or Rayleigh's approach?*

As noted, a single communication involving all fields and all radii could be performed. In the early stages of the project we have performed tests using such a strategy. This has the following consequences:

(1) allows much large message sizes

(2) fewer synchronization points are needed

(3) memory requirement grows with the number of radial points (since all fields and buffers for each radial point must be stored imultaneously for a single communication ).

Our Figure 3 shows that "packing" more messages past 1,750KiB (queue of length 9) does not provide any visible performance benefit for the hardware we used in our tests, voiding (1). Concerning (2), some of the early experiments showed an equivalent or slightly superior performance of the queue algorithm when many radial points were used. Moreover, for simulations involving many radial points, (3) is exacerbated.

We agree that the combining all messages in a single communication could be useful in several scenarios and we may revisit this part of MagIC in future releases, enabling the user to choose between strategies.

**Referee#1, Comment 3:** *And, SHTns is used in the present study. I wonder if the authors calculate the Legendre polynomials at Gauss-Legendre points in the initialization, or alculate during each Legendre transforms. I remember that SHTns has both feature, so it would be helpful which approach is chosen and why the authors choose one.*

The "on-the-fly" strategy is used. This is the recommendation of SHTns' author for $\ell_{max} > 32$ (see (Schaeffer, 2013) for more details).

**Referee#1, Comment 4:** *Lastly, the authors defined Tt. However, I can't find any information for Tt in Table 3. Can I find the data from the other table? So, I lost a direction to figure out the following discussion using Tt in page 26. Please provide how to figure out Tt.*

$T_t$ and $t_t$ are given in Table 2 under the column "time". In the revised text, we added these columns to Table 3 as well.

**Comment 5:**Referee#1, *And, these are some minor suggestions: In line 2, can 'magnetohydrodynamics' be one word?*

We modified all occurrences throughout the text to "magnetohydrodynamics".

**Referee#1, Comment 6:** *In line 5, I think "parallelization" would be more explicit than "implementation".*

We changed all occurrences of "hybrid implementation" to "hybrid parallelization" throughout the text, but we kept the term "1d-hybrid implementation", since it refers to a "version" of the code. We hope that this change meets the expectations.

**Referee#1, Comment 7:** *In line 27, "mag" would be capitalized as "MAG" or "Mag"*

This has been corrected as requested.

**Referee#1, Comment 8:** *In line 65, Full name should be represented for "Non-Uniform Memory Access (NUMA)" first.*

This has been modified, but on line 59 where it appears first.

**Referee#1, Comment 9:** *In line 117, It would be better to add dimensionless "self" gravity.*

This has been modified as requested.

**Referee#1, Comment 10:** *In equation (14) and (15), the diffusion term appears in the left hand side and right hand side, respectively. I would like to show this term in the same side in the both equations.*

This has been modified as requested.

**Referee#1, Comment 11:** *In line 202, I would like to show the equation using $M_l^{\delta t}$ and $B_{lm}^i$ below of the equation (17). I looked for the definition of $M_l^{\delta t}$ and $B_{lm}^i$ for a while.*

We added a sentence in the text to clarify the definition of $M_l^{\delta t}$ and $B_{lm}^i$.

**Referee#1, Comment 12:** *in line 329 and 330, I prefer to say "component" instead of "field", if the scalar "field" includes toroidal and toroidal components for vectors.*

We fear that using the word "components" may erroneously lead the reader to believe that we are solving for vector components. To avoid this confusion we prefer to refer to them "scalar fields".

**Referee#2, Comment 1:** *Considering that a large portion of todays supercomputers include accelerators, possibly making up most of their raw computational power, does the 2D strategy bring any advantages with regards to using accelerators?*

This is a very valid point. Unfortunately, porting a large production code like MagIC to GPUs (or more generally speaking to a discrete "accelerator") is a significant challenge which is way beyond the scope of this work. Currently, and at least for the next couple of years, the community using MagIC has access to large-scale CPU (x86_64 CPUs from Intel or AMD) resources, e.g. in Germany or France, justifying the development efforts for "modernizing" the CPU-only version of MagIC.

That said, we do fully agree with the referee that porting to GPUs eventually might even become unavoidable in the light of the current hardware trends and technological developments in HPC. Realistically, a GPU port of MagIC would start out from the OpenMP parallelization which is already there for the 1D-MPI parallelization (which is envisaged also for the 2D-MPI parallelization). In that sense, the newly developed 2D MPI-strategy does not bring immediate advantages with regards to using accelerators but it would add flexibility to a future GPU version of MagIC in the same way as it now does for the CPU-only version. We have added a short paragraph at the end of Section 5, Conclusions and Future Work (lines 601–606 of the revised manuscript) which addresses this point and sketches a conceivable GPU-porting strategy for MagIC.

**Referee#2, Comment 2:** *The discussion of the transposition in section 3.3 is confusing. The paragraph starting at line 335 discusses the importance of the size of the queue, but the MPI algorithms (l. 325) specify a MPI communication*

"per scalar field". The discussion in Section 4.1 also seems to imply a single communication call per "queue". Can you clarify this?

Indeed, the description in lines 325 was wrong. The algorithm performs one communication for all queued fields. The text has been updated accordingly.

**Referee#2, Comment 3:** *The performance benchmarks provide insight into the behaviour of the 2D parallelization implementation but are executed on a single cluster. A more general discussion on what kind of performance to expect on another HPC cluster depending on its technical characteristics would be helpful for the wider community.*

Indeed such a discussion was missing from the main manuscript. We added a paragraph in in Section 4.5 of the manuscript (lines 522–528), elaborating on how these performances should translate into other clusters.

**Referee#2, Comment 4:** *The strong scaling experiment with the 1D hybrid strategy increases the number of threads for runs with higher number of cores. This is counter intuitive as it leads to even smaller computational load per thread which I would expect to affect the scaling negatively. Can you explain this behaviour?*

We performed tests with the 1d-hybrid code with 10 and 20 threads, but always with the same total number of threads. For instance, the test with 1,000 cores was executed using 100 MPI ranks and 10 threads per rank, or 50 MPI ranks and 20 threads per rank. In both scenarios, the workload per thread is the same. What changes is the amount of data being handled by each MPI rank. The transition from 10 threads to 20 threads shows that, at some point it is better for MPI to handle larger chunks of data, which is expected.

**Referee#2, Comment 5:** *At high resolution, the memory footprint of a pseudo-spectral code can become important. Is there a benefit, at the memory level, to use a 2D distribution in MagIC?*

This is indeed another benefit of the 2d-MPI implementation. Since it allows the use of more resources, the user may have access to more (distributed) memory as well. We added a comment in the revised version of the text, in Section 4.5, lines 516–517.

It may be noted that the proposed 2D MPI distribution requires additional memory for storing the queued fields during the $\theta$-transposition. However, in section 3.3 we clarify that the queue size (and thus, extra memory requirement) can be controlled by the user as well. The benefit of having more distributed memory available can easily overcome the disadvantage of having to store a controllable extra number of scalar fields.

**Referee#2, Comment 6:** *l. 132: Plm should be the "associated Legendre polynomials" and not the "Legendre polynomials".*

This has been corrected as requested.

**Referee#2, Comment 7:** *Table 3: Adding a $\kappa Tr$ column would make it easier to follow the discussion at the end of Section 4.6*

This has been updated as suggested.

Please, let us know if any of the changes are insufficient or if they could be improved. We look forward to hearing from you.

Sincerely,
Rafael Lago